# Are Voltage Sensors Really Embedded in Muscarinic Receptors?

**DOI:** 10.3390/ijms24087538

**Published:** 2023-04-19

**Authors:** Malka Cohen-Armon

**Affiliations:** The Sackler School of Medicine, Department of Physiology and Pharmacology, and Sagol School of Neuroscience, Tel-Aviv University, Tel-Aviv 69978, Israel; marmon@tauex.tau.ac.il

**Keywords:** muscarinic receptors, voltage-dependent sodium channels, synaptoneurosomes, Go-proteins, voltage-dependent muscarinic receptors’ signaling

## Abstract

Unexpectedly, the affinity of the seven-transmembrane muscarinic acetylcholine receptors for their agonists is modulated by membrane depolarization. Recent reports attribute this characteristic to an embedded charge movement in the muscarinic receptor, acting as a voltage sensor. However, this explanation is inconsistent with the results of experiments measuring acetylcholine binding to muscarinic receptors in brain synaptoneurosomes. According to these results, the gating of the voltage-dependent sodium channel (VDSC) acts as the voltage sensor, generating activation of Go-proteins in response to membrane depolarization, and this modulates the affinity of muscarinic receptors for their cholinergic agonists.

## 1. Introduction

Muscarinic receptors are seven transmembrane cholinergic receptors. They possess an extracellular amino-terminus, seven membrane-traversing hydrophobic helices, and a cytoplasmic carboxyl terminus [1]. Agonists and antagonists binding to these receptors affect downstream signal transduction mechanisms, mainly by manipulating ion currents and via activation of heterotrimeric GTP-binding proteins [1,2].

Muscarinic receptors are modified by post-translational modifications affecting their binding to other proteins and lipids in the cell membrane [1]. In addition, the affinity of muscarinic receptors for their cholinergic agonists is modulated by membrane depolarization [1,2]. Recently, this characteristic has been attributed to a depolarization-induced charge movement in the muscarinic receptor. However, this explanation is inconsistent with numerous findings that are briefly presented here. Unveiling the molecular mechanism underlying the voltage-dependent affinity modulation of muscarinic receptors for their agonists is of significant importance for targeting the physiological outcome of muscarinic receptor stimulation in a variety of signal transduction mechanisms [3].

## 2. Discussion

Different affinities of muscarinic receptors for their agonists were identified four decades ago [4]. A depolarization-induced modulation of the affinity of muscarinic receptors for their cholinergic agonists was first identified in the rat brain cortex, brain stem, and atria [5]. It was further confirmed for the affinity of the M2 muscarinic receptor subtype [6]. The voltage-induced modulation of the muscarinic receptors affinity for agonists is apparently one of the common features of muscarinic receptor subtypes [4,5,7]. Recent studies attribute the voltage-dependent modulation of muscarinic receptor affinity to a voltage-induced charge movement in the muscarinic receptor [6,8]. A charge movement producing gating current was attributed to modifications in the intramolecular loops L2 and L3 of muscarinic receptors, which are implicated in the coupling of the muscarinic receptor to trimetric G-proteins [6,8]. Unlike previous reports, these studies attribute opposite effects of membrane depolarization on the affinity of muscarinic receptor subtypes M1 and M2 [5,8]. M1 is the main muscarinic receptor subtype in the brain, and the M2 muscarinic receptor subtype is most abundant in the atria [1]. A voltage-induced charge movement in the muscarinic receptors was attributed to charged residues in the L3 intramolecular loop of the muscarinic receptor [6,8]. However, this idea was inconsistent with the affinity changes resulting from their removal [9]. Removal of charged residues from the amino-terminal end of loop L3 in M2 muscarinic receptors abolished their depolarization-induced modification [9], but did not abolish the associated gating current measured in response to membrane depolarization, as expected for a gating sensor [9]. Moreover, gating currents were measured in the M1 subtype of muscarinic receptors in response to membrane depolarization, despite the fact that M1 receptors lack the charged sequence in loop L3 [9]. Thus, the conclusion of a tight correlation between the binding of muscarinic receptors to trimetric G-proteins and their affinity modulation was accurate [5,9], but it did not indicate any tight correlation between a voltage-dependent charge movement in muscarinic receptors and the voltage-dependent modulation of their affinity for agonists [9,10,11].

A different mechanism may underlie the effect of membrane potential on the affinity of muscarinic receptors. This mechanism accurately fits findings indicating the indispensable role of the voltage-dependent sodium channel (VDSC) gating (even when Na^+^ entry is blocked) in the depolarization-induced modulation of the muscarinic receptor affinity for cholinergic agonists [12,13]. According to these findings, depolarization-induced opening of VDSC is essential for a depolarization-induced activation of trimetric Go-proteins, which modulate the affinity of muscarinic receptors for their cholinergic agonists from high affinity to low affinity [12,13,14]. These findings are briefly described.

An interaction between VDSC and muscarinic receptors was identified when muscarinic cholinergic agonists induced the opening of VDSC in synaptoneurosomes prepared from the brain cortex or brain stem [15]. Synaptoneurosomes consist of a presynaptic, resealed neuron (synaptosomes) attached to a resealed postsynaptic neuron (neurosomes), both containing the original content of their cytoplasm. Synaptoneurosomes were first prepared and examined by the group of Creveling [16].

The effect of membrane potential on different receptors in synapses was examined in synaptoneurosomes [5,12,13,14,15]. In these experiments, binding of cholinergic agonists to muscarinic receptors induced a Tetrodotoxin (TTX)-sensitive ^22^Na^+^ uptake, which was blocked by the muscarinic receptor antagonist atropine (Ref. [15]; Figure 1). In addition, specific binding of toxins to sites in the open configuration of VDSC, including the specific binding of labeled Batrachotoxin ([^3^H]BTX), was induced at resting membrane potential by the binding of cholinergic agonists to muscarinic receptors in the synaptoneurosomes. Their displacement by atropine prevented the induced [^3^H]BTX binding by muscarinic agonists (Refs. [13,15]; Figure 1).

The binding of certain toxins to the open configuration of VDSC either keeps the VDSC in their open configuration (e.g., Batrachotoxin (BTX) and the S-enantiomer of the cardio-tonic drug DPI) or prevents the opening of VDSC (e.g., the R-enantiomer of the cardio-tonic drug DPI) [13,17,18,19,20,21].

Muscarinic cholinergic receptor agonists dose-dependently enhanced the specific binding of [^3^H]BTX to open VDSC in the synaptoneurosomes at resting membrane potential, while Na^+^ entry was blocked (Refs. [13,15]; Figure 1). Thus, these findings suggest that binding of cholinergic receptor agonists to muscarinic receptors induces the opening of VDSC at resting membrane potential and that the ^22^Na^+^ entry is blocked by TTX, which blocks sodium currents via VDSC in the brain (Ref. [15]; Figure 1).

A reciprocal effect of the VDSC gating on the affinity of muscarinic receptors for their agonists was identified as well [12,13]. The high affinity binding of [^3^H]acetylcholine to muscarinic receptors in the brain synaptoneurosomes was substantially reduced under membrane depolarization [12,13], and the open configuration of VDSC was indispensable for this voltage-induced affinity change in the muscarinic receptors, even when Na^+^ entry was blocked [12,13,14]. Thus, a depolarization-induced reduction in the high-affinity binding of [^3^H]acetylcholine to muscarinic receptors was not measured when the opening of VDSC was prevented by the binding of the R-enantiomer of the cardiotonic drug DPI to the VDSC [12,13,14,20] (Figure 2). According to these results, the open configuration of the VDSC (but not the Na^+^ current) was indispensable for the depolarization-induced modulation of the muscarinic receptors’ affinity for cholinergic agonists, from a high affinity to a low affinity (Refs. [12,13,14]; Figure 2). The binding of antagonists to muscarinic receptors was not similarly affected by membrane depolarization or by the open configuration of the VDSC [5,12,13].

Many findings identified the involvement of muscarinic receptor coupled trimetric G-proteins in the depolarization-induced affinity modulation of muscarinic receptors [5,8,9,12,14,22]. One of these results showed that the L2 and L3 loops in the muscarinic receptor are implicated both in the receptor coupling to trimetric G-proteins and in the depolarization-induced modulation of its affinity for agonists [8,9]. These findings are in accordance with the indispensable role of the depolarization-induced activation of trimetric Go-proteins in the depolarization-induced modulation of the muscarinic receptors’ affinity for their cholinergic receptor agonists [14].

The pertussis toxin (PTX)-sensitive trimetric Gi- and Go-proteins are activated by muscarinic agonists [1,2,5,12,14]. The activation of these G-proteins was measured in-situ in synaptoneurosomes by the covalent binding of a labeled GTP [14]. The covalent binding of labeled GTP to trimetric Go-proteins was induced by the binding of cholinergic receptor agonists to muscarinic receptors as well as in response to membrane depolarization [14].

Go-proteins are a highly conserved subtype of trimetric G-proteins, expressed in excitable tissues of numerous species [23]. Go-proteins are highly expressed in several regions of the mammalian brain and in peripheral nervous tissues. In addition, Go-proteins are highly expressed in the cardiac atria [23]. Unlike Gi-proteins, Go-proteins are activated in response to membrane depolarization, even in the absence of receptor stimulation [14]. Activation of Go-proteins was measured in-situ by tracing the covalent binding of [α^32^P]GTP-azidoanilide under UV-irradiation in depolarized synaptoneurosomes [14]. No covalent binding of [α^32^P]GTP-azidoanilide to Go proteins was measured at resting membrane potential [14]. Furthermore, [α^32^P]GTP-azidoanilide binding to Go-proteins in the depolarized synaptoneurosomes was not dependent on depolarization-induced stimulation of muscarinic receptors. Namely, [α^32^P]GTP-azidoanilide binding to Go-proteins in response to membrane depolarization was not abolished in the presence of antagonists of muscarinic cholinergic receptors. Neither was the depolarization-induced [α^32^P]GTP-azidoanilide binding to Go-proteins prevented by antagonists to additional receptors, including adrenergic, dopaminergic, and serotonergic receptors [14]. In contrast, the open configuration of VDSC in the depolarized synaptoneurosomes was indispensable for the depolarization-induced exchange of GDP by [α^32^P]GTP-azidoanilide in Go-proteins, even when Na^+^ entry was blocked by TTX [14]. In addition, the α-subunit of Go-proteins co-immunoprecipitated with the α-subunit of the VDSC in depolarized synaptoneurosomes [14], indicating a possible interaction between these proteins in the depolarized membranes [14].

Furthermore, the displacement of GDP by GTP in Go-proteins in response to membrane depolarization was accompanied by a depolarization-induced modulation of the high affinity binding of [^3^H]acetylcholine to muscarinic receptors from a high to a low affinity [14]. In addition, modifications preventing the activation of PTX-sensitive trimetric G-proteins interfered with the depolarization-induced activation of Go-proteins and with the depolarization-induced modulation of muscarinic receptors affinity for their cholinergic agonists [5,12,13,14]. Thus, PTX-induced ADP-ribosylation of the α-subunit of Go- and Gi-proteins, as well as the irreversible binding of GDPβS to trimetric G-proteins, prevented the activation of Go-proteins by muscarinic agonists as well as their activation by membrane depolarization [12,14]. Furthermore, these modifications in G-proteins also prevented the depolarization-induced modulation of the high affinity of muscarinic receptors for [^3^H]acetylcholine into a low affinity [12,14].

These findings associate the depolarization-induced exchange of GDP by [α^32^P]GTP-azidoanilide in activated Go-proteins with the depolarization-induced affinity modulation of muscarinic receptors for [^3^H]acetylcholine [12,14].

Furthermore, the open configuration of VDSC was a pre-requisite for the effect of G-protein activation on muscarinic affinity modulation (even when Na^+^ entry was blocked) [12,13,14]. Thus, the effect of a permanent binding of GTP to G-proteins (Gpp(NH)p) keeping G-proteins activated on the affinity modulation of muscarinic receptors for [^3^H]acetylcholine was eliminated when the opening of VDSC was prevented [14]. A permanent binding of Gpp(NH)p reduced the high affinity of the muscarinic receptor for [^3^H]acetylcholine, but only when the VDSC were kept in their open configuration in synaptoneurosomes treated with BTX or the S-enantiomer of DPI (Refs. [12,13,14]; Figure 3). Preventing the open configuration of the VDSC by the binding of the R-enantiomer of DPI to the VDSC [20] prevented the effect of persistent activation of G-proteins on the high- to low-affinity modulation of muscarinic receptors for [^3^H]acetylcholine (Ref. [14]; Figure 3). According to these results, the open configuration of the VDSC was required for affinity modulation of muscarinic receptors by activation of G-proteins (Figure 3).

Thus, since the affinity of muscarinic receptors for agonists is modulated by activation of PTX-sensitive Gi- and Go-proteins [14], and Go-proteins are activated by membrane depolarization, membrane-depolarization-induced activation of Go-proteins could modulate the affinity of muscarinic receptors for agonists (Figure 2 and Figure 3).

Furthermore, the depolarization-induced opening of VDSC was indispensable for the depolarization-induced activation of Go-protein as well as for the depolarization-induced affinity modulation of muscarinic receptors (Refs. [12,13,14]; Figure 2 and Figure 3).

Thus, according to these results, VDSC gating [17,18,19,20,21], acting as the voltage sensor generating activation of Go-proteins in response to membrane depolarization, causes the affinity modulation of the muscarinic receptor for cholinergic agonists in response to membrane depolarization.

This mechanism is in accordance with the suggested role of loops L2 and L3 in the muscarinic receptor, which are implicated in the coupling of the muscarinic receptor with G-proteins and in the depolarization-induced affinity modulation of the muscarinic receptors [8,9,22]. Moreover, the indispensable role of the open configuration of VDSC in the depolarization-induced affinity modulation of muscarinic receptors for their cholinergic agonists [12,13,14] may explain the discrepancy between the measured gating current even when charges in loop L3 were missing and the attributed gating current to charge movement in the L3 loop of muscarinic receptors [9].

So, in addition to other identified inconsistencies [9,11], an embedded charge movement in the muscarinic receptor [6,8] does not seem to be in line with findings indicating that the depolarization-induced modifications in VDSC are indispensable for depolarization-induced modulation of the muscarinic receptor affinity for their cholinergic agonists (Refs. [12,13,14]; Figure 2 and Figure 3).

Notably, modifications in G-proteins preventing their activation did not interfere with muscarinic cholinergic agonists inducing the opening of VDSC at resting membrane potential [15]. This result may exclude the involvement of G-proteins’ activation in the opening of VDSC by cholinergic agonists that bind with a high affinity to muscarinic receptors at resting membrane potential [12,13,14]. Thus, under physiological conditions, muscarinic agonists inducing the opening of VDSC at resting potential could promote repetitive firing, as identified in pacemakers’ activity [24,25]. Depolarization-induced modulation of the muscarinic receptors’ affinity for agonists from high to low affinity (Refs. [5,12,13,14], Figure 2) has been associated with signal transduction mechanisms mediated by activation of G-proteins, phospholipase-C (PLC) activation, *inositol* 1,4,5-trisphosphate (IP_3_)-induced Ca^2+^-release from intracellular stores [1], modulation of ion currents [1,2], and activation of phosphorylation cascades [1].

In summary, a mechanism suggesting that the affinity of muscarinic receptors for their cholinergic agonists is modulated by membrane depolarization due to a voltage-induced charge movement in the muscarinic receptor is inconsistent with findings indicating the indispensable role of the VDSC gating in the depolarization-induced modulation of the muscarinic receptor’s affinity for cholinergic agonists [12,13,14].

## Figures and Tables

**Figure 1 ijms-24-07538-f001:**
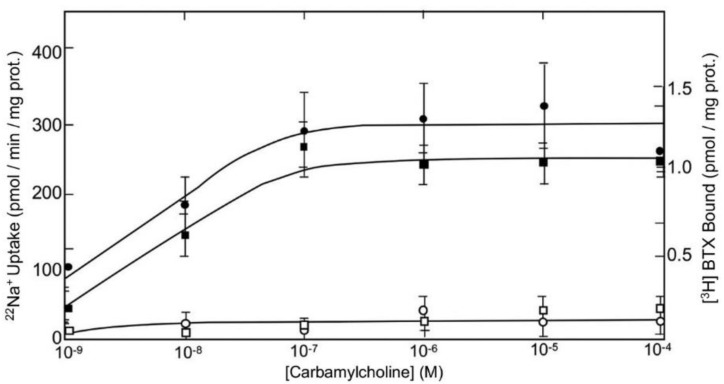
TTX-blockable ^22^Na^+^ uptake (circles) and [^3^H]BTX binding (squares) to pertussis toxin-treated synaptoneurosomes, as a function of carbamylcholine concentration in the presence (
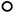
, 
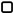
) and absence (
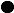
, 
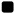
) of atropine (0.1 µM). The specific carbamylcholine-induced binding of [^3^H]BTX was measured in the presence of TTX (1 µM). The non-specific carbamylcholine-induced ^22^Na^+^ uptake, both in the presence and absence of atropine, was measured in the presence of TTX (1 µM) [15].

**Figure 2 ijms-24-07538-f002:**
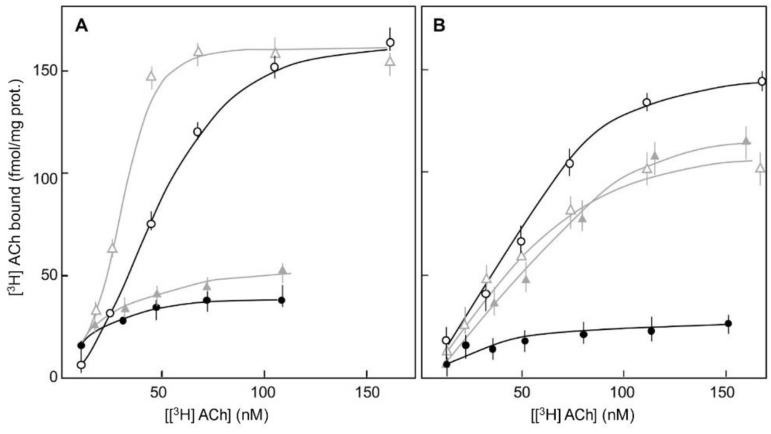
Effects of DPI-enantiomers on the high affinity binding of [^3^H]acethylcholine ([^3^H]ACh) to muscarinic receptors in rat brain synaptoneurosomes at resting membrane potential (
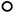
, 
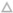
) and upon depolarization (
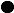
, 
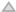
). [^3^H]Ach binding measured in untreated synaptoneurosomes (black lines, circles) and in synaptoneurosomes treated (grey lines, triangles) with the S-enantiomer of DPI (5 µM) in the presence of 1 µM TTX (**A**) or with the R-enantiomer of DPI (5 µM) (**B**), are presented. The non-specific binding of [^3^H]ACh was measured in the presence of 1 µM atropine. (Ref. [12]).

**Figure 3 ijms-24-07538-f003:**
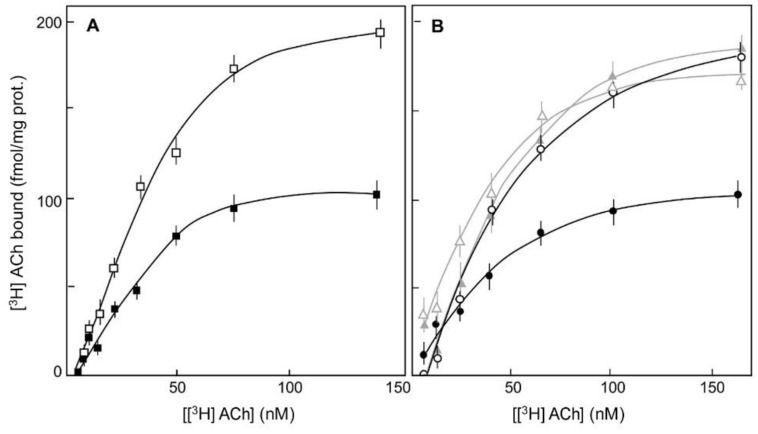
The effect of a permanent binding of GTP (Gpp(NH)p) (200 μM) on the high-affinity binding of [^3^H] acetylcholine ([^3^H] ACh) to muscarinic receptors in rat brainstem synaptoneurosomes is dependent on the configuration of VDSC in the presence of the R-enantiomer or the racemic mixture of the cardiotonic drug DPI. (**A**). [^3^H] ACh binding to muscarinic receptors when G-proteins bind Gpp(NH)p (
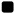
) and in the absence of Gpp(NH)p (
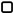
). (**B**). [^3^H] ACh binding was measured after treatment with the DPI R-enantiomer (5 μM) (grey curves, triangles) or with the racemic mixture of the cardio-tonic drug DPI (5 μM) (black curves, circles) in the presence (
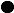
, 
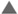
) and in the absence of Gpp(NH)p (
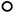
, 
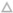
). The non-specific binding of [^3^H]ACh was measured in the presence of 1 μM atropine. (Ref. [12]).

## Data Availability

Not applicable.

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
