# Peer review of "Are Voltage Sensors Really Embedded in Muscarinic Receptors?"

_ijms, 2023, doi:10.3390/ijms24087538_

Round 1
Reviewer 1 Report
Malka gives an opinion against the idea that the muscarinic acetylcholine receptors themselves have voltage sensors embedded. Although this idea appears to be interesting, this manuscript is very rough in writing and is not written in a way that makes it easy for the reader to understand. Some statements are scientifically incorrect. There are many points that should be addressed and may serve to amend this manuscript, as follows:
1. Abstract: “muscarinic receptors” should be “muscarinic acetylcholine receptors”. “revealed” should be “suggested”, because the author’s idea appears to be based on the results (Figs. 1-3) of binding experiments in synaptoneurosomes. This point should be stated in Abstract.
2. Keywords: “muscarinic signaling” should be “muscarinic receptor signaling”. The words should be unified in style.
3. Introduction: “affect” should be “affects”.
4. Second paragraph on page 2: not “nerosomes” but “neurosomes”? TTX should be defined where it first appears. Is “22Na+ currents” OK? 22Na+ influx (entry)? It seems that Na+ currents cannot be recorded from synaptoneurosomes. The “muscarinic antagonist” and “labeled toxin” should be written specifically. “[3H]BTX” should be put following “labeled toxin”. “resting potential” should be “resting membrane potential” throughout the manuscript. Not “keep” and “loch-down” but “keeps” and “loch-downs”.
5. First paragraph on page 3: is “even when Na+ current was blocked (14; Figure 1)” OK? Was the data shown in Fig. 1 obtained in the presence of TTX? If so, this point should be stated clearly. This may be known by referring to (14), but it should be written so that it can be understood without reference to (14). Such a problem applies to the entire manuscript. Is “Na+ (sodium) current” OK? (see comment 4)
6. Second paragraph on page 3: is “muscarinic agonists” in line 2 OK? Not “Na+ current” but “Na+ entry”?
7. Third paragraph on page 3: “.. muscarinic coupled ..” should be revised.
8. Fourth paragraph on page 3: “pertussis toxin” should be defined as “PTX” (see page 4).
9. Fifth paragraph on page 3: is “[α32P]GTP(azidoanilide GTP)” OK? [α32P]GTP azidoanilide? “Na+ current” should be “Na+ channel”.
10. Second paragraph on page 4: not “preventing” but “prevented”?
11. Third paragraph on page 4: in line 5 from the bottom, “(Gpp(NH)p)” should be “Gpp(NH)p”.
12. Fifth paragraph on page 4: not “Na+ current” but “Na+ entry”? Not “high-to low” but “high-to-low”? (see third paragraph on page 4)
13. Sixth paragraph on page 4: “.. muscarinic induced ..” should be revised.
14. First paragraph on page 5: “.. muscarinic -induced ..” should be revised. Not “Na+ current” but “Na+ entry”? “PLC” should be expanded. “Ca” should be “Ca2+”.
15. Figs. 1-3: please use either “protein” (Fig. 1) or “prot.” (Figs. 2 and 3). Not “pmoles” but “pmol” in Fig. 1?, if “fmol” is used in Figs. 2 and 3. Please check this point.
16. The legend of Fig. 1: it will be better to put “(squares)” after “binding”. Please put a period at the end of this legend.
17. Fig. 2 and this legend: “ACh” should be used instead of “AcCh”. Please put “membrane” before “potential” and “depolarization”. The expression of “[K+] buffer” is scientifically incorrect. “Black triangle” should be “grey triangle” (see Fig. 2).
18. The legend of Fig. 3: please use either “cardiotonic drug DPI” or “cardic-tonic drug DPI” throughout this legend and the text.
19. References; the style of references should be uniform. The titles of refs. 7 and 8 are different in style; the journal names of refs. 10 and 19 are different in style.
20. There seem to be much more mistakes than pointed out above. Please check your manuscript very carefully.
Author Response
Replies:
The manuscript has been carefully checked and revised according to your suggestion (comment 20). I have addressed all your comments 1-15,19 in the revised manuscript (labeled by track changes).
The abbreviations in the figures and the legends in the revised version were changed according to your comments 16-18. The references style is now uniform according to your comment 19 (labeled with track changes). My reply to comment 5 is included in the revised legend of figure 1.
Comment 5:
First paragraph on page 3: is “
even when Na
current was blocked (14; Figure 1)” OK?
Was the data shown in Fig. 1 obtained in the presence of TTX?
If so, this point should be stated clearly.
This may be known by referring to (14), but it should be written so that it can be understood without reference to (14).
Reply:
Yes. In Figure 1, The specific carbamylcholine-induced binding of [3H]BTX was measured in the presence of TTX (1 mM). The non-specific carbamylcholine-induced 22Na+ uptake, both in the presence and in the absence of atropine, was measured in the presence of TTX (1 mM) (revised legend). Ref 14 is ref 15 in the revised version.

Reviewer 2 Report
This report by Cohen-Armon Malka summarizes differences in the affinity of muscarinic receptors for agonists. So far, it has been mainly discussed in relation to the voltage-dependent sodium channel, but many have pointed out the flaws in that way of thinking.
However, this report does not mention new possibilities for different affinities to agonists. The author should show the mechanism.
In particular, about "These findings revealed a more complicated mechanism in which, the open configuration of VDSC and activation of Go-proteins are indispensable for the depolarization-induced affinity modulation of muscarinic receptors." in abstract, please suggest a part of the more complicated mechanism that you think.
Author Response
-
I am not aware of papers contradicting the implication of VDSC gating in the voltage-dependent affinity of muscarinic receptors for agonists. In Rezenfeld et al., 2021, sodium inward current was blocked, but the effect of depolarization-induced modifications in VDSC gating on the depolarization-induced affinity modulation of muscarinic receptors was not examined, nor excluded.
2. Based on the presented data, the suggested mechanism underlying the depolarization-induced changes of the muscarinic receptors affinity for their agonists is included in the abstract and page 4 ( revised version).
The presented results are consistent with a mechanism by which voltage induced modification of VDSC gating generates activation of Go-proteins in response to membrane depolarization (14). The depolarization-induced Go-proteins activation modulates the muscarinic receptors affinity for their cholinergic agonists (12-14).

Round 2
Reviewer 1 Report
This revised manuscript has been largely amended according to my comments, and there is no concern in this manuscript except for the following minor comments:
1. Line 14 in the second paragraph on page 2: “.. subtype are most ..” should be “.. subtype is most ..”.
2. Line 21 in the second paragraph on page 2: “.. membrane- depolarization ..” should be “.. membrane depolarization ..”.
3. First line on page 3: “.. muscarinic antagonist ..” should be “.. muscarinic receptor antagonist ..”.
4. Line 5 on page 3: please delete “the muscarinic antagonist,” (please see first line on page 3).
5. Line 9 on page 3: not “(13,17-21),” but “(13,17-21)),”? Please check this point.
6. Line 1 in the third paragraph on page 3: “cholinergic agonists” should be “cholinergic receptor agonists”. Although not specifically pointed out, similar mistakes can be found elsewhere.
7. Line 4 in the third paragraph on page 3: “induce” should be “induces”.
8. First line in the fifth paragraph on page 3: “muscarinic coupled” should be “muscarinic receptor coupled”.
9. Line 7 from the bottom on page 3: “in-“ should be “in”.
10. Line 8 on page 4: “Go proteins” should be “Go-proteins”.
11. Line 8 in the second paragraph on page 4: “Go and Gi-proteins” should be “Go- and Gi-proteins”.
12. Line 18 from the bottom on page 4: “high- to low affinity” should be “high- to low-affinity”.
13. Fig. 1: “pmol.” on the vertical axis of this figure should be “pmol”. (see the vertical axes in Figs. 2 and 3)
14. References: the titles of Refs. 1, 8 and 25 are different in style from the others. Please amend this point.
15. This manuscript should be checked very carefully once again.
Author Response
Dear Reviewer,
Thank you for the minor comments. I have addressed all of them in the manuscript (with track changes).
Reviewer 2 Report
Overall well revised. No more opinions. I think it's fine to accept.
Author Response
Thank you for accepting the manuscript.